# Synthesis of Benzoxazine-Based N-Doped Mesoporous Carbons as High-Performance Electrode Materials

**Haihan Zhang, Li Xu * and Guoji Liu ***

School of Chemical Engineering and Energy, Zhengzhou University, Zhengzhou 450001, China; 15736873061@163.com

* Correspondence: xuli@zzu.edu.cn (L.X.); guojiliu@zzu.edu.cn (G.L.)

**Abstract:** In this work, nitrogen-doped carbon materials (NCMs) were prepared using aniline-phenol benzoxazine (BOZ) or aniline-cardanol benzoxazine as the carbon precursor and SBA-15 as the hard template. The effects of the carbonization temperature (700, 800, and 900 °C) and different nitrogen contents on the electrochemical properties of carbon materials were investigated. The samples synthesized using aniline-phenol benzoxazine as precursors and treated at 900 °C (NCM-900) exhibited an excellent electrochemical performance. The specific capacitance was 460 F/g at a current density of 0.25 A/g and the cycle stability was excellent (96.1% retention rate of the initial capacitance after 2000 cycles) in a 0.5 M $H_2SO_4$ electrolyte with a three-electrode system. Furthermore, NCM-900 also exhibited a high specific capacitance, comparable energy/power densities, and excellent cycling stability using a symmetrical electrode system. The characterization of the morphology and structure of the materials suggests it possessed an ordered mesoporous structure and a large specific surface area. NCM-900 could thus be considered a promising electrode material for supercapacitors.

**Keywords:** benzoxazine; nitrogen-doped mesoporous carbon; electrode material; electrochemical performance; supercapacitor

---

## 1. Introduction

As a type of energy-storage device, supercapacitors offer a promising solution for the fast storage the excess electrical energy [1,2]. Electrochemical double-layer capacitors (EDLCs) or supercapacitors have recently attracted the attention of many researchers due to their considerable capacity, high power density, wide operating temperature range, short charging time, and long cycle life [3–6]. Energy is stored in supercapacitors via the electrostatic accumulation of charge at the electrode surface. As the most commonly used electrode materials for supercapacitors, carbon materials show excellent electrical and thermal conductivity, a large specific surface area, and high chemical stability [7,8].

The limited specific capacity of carbon materials restricts its application as a good electrode material. This is shown to be ameliorated by doping heteroatoms, such as nitrogen (N), boron (B), phosphorus (P), and sulfur (S), into the carbon structures [9,10]. Doped heteroatoms in carbon materials can effectively enhance the specific capacitance by adding pseudocapacitance, which arises from the redox reaction of the heteroatoms [11,12]. Furthermore, doping could improve the wettability between the electrolyte and the electrode material [13]. Among the doped carbons, nitrogen-doped carbon materials (NCMs) have been shown to be promising at improving the capacitance via surface faradaic reactions without sacrificing the high rate capability and long cycle life. The new NCMs show good electronic conductivity and they can be easily prepared at low cost. To our knowledge, nitrogen has become the most-studied doped heteroatom for carbonaceous electrode materials [14,15].

Two main methods for the preparation of heteroatom-doped carbon materials have been developed so far: in situ doping using nitrogen-containing precursors (such as urea-polymer [16], polyaniline (PANI) [17], cyanamide [18], dicyandiamide [19], melamine [20]) and the post-treatment of porous carbons (using $NH_3$ or amines). In in situ doping, heteroatoms usually enter directly into the skeleton of the carbon materials. This method could resolve the problem of nitrogen-containing functional groups on the surface of the porous carbons, which decrease the available surface area and pore volume [21,22]. In our investigation, the heteroatom was introduced using benzoxazine as a carbon precursor. The benzoxazine-based porous carbon material introduces various organic elements, such as oxygen and nitrogen [23–25]. Meanwhile, the high surface area and ordered porous structure of electrode materials are also important factors that affect the performance of supercapacitors. The nanostructured porous materials synthesized using template mesoporous silica (e.g., SBA-15, MCM-41) usually meet the above material requirements. In this method, the carbon material can be obtained inside the pores of the silica template, which can easily be removed by the treatment of hydrofluoric acid or sodium hydroxide [26,27]. The recovered carbon material, which keeps the morphology of the silica host, has a controlled pore size and ordered porous structure. Thus, the application of templating agents (e.g., SBA-15, MCM-41) provides a good idea for the design of electrode materials with a controllable morphology [28]. To date, benzoxazine-based, heteroatom-doped carbon materials that are synthesized using the hard template have not been well studied.

In this paper, for the first time, we synthesized nitrogen-doped mesoporous carbons with two different precursors (aniline-phenol benzoxazine and aniline-cardanol benzoxazine). Through the hard template (SBA-15) synthetic method, we investigated the effect of these different phenol sources on the nitrogen content and the effect of the carbonization temperature on the structure of the carbon materials. The electrochemical properties were also investigated. Due to having almost the same microscopic morphology and the same type of heteroatom, the two carbon materials provided analogous superior electrochemical performance. However, the carbon materials with the higher nitrogen content (NCM, those produced using an aniline-phenol benzoxazine precursor) displayed better electrochemical properties than those with the lower nitrogen content (ACNCM, aniline-cardanol benzoxazine precursor).

## 2. Experimental Section

### 2.1. Materials

Aniline, phenol, cardanol, formaldehyde solution (37 wt%), toluene, tetrahydrofuran, anhydrous ether, and hydrofluoric acid were all of analytical grade and were purchased from Tianjin Comomi (Comomi, Hebei District, Tianjin, China); acetylene black and SBA-15 were purchased from Pioneer Nano (Pioneer Nano, Gulou District, Nanjing, China); and Nafion perfluorinated resin solution was purchased from Aldrich (Aldrich, Pudong District, Shanghai, China).

### 2.2. Synthesis of Aniline-Phenol Benzoxazine

Benzoxazine was synthesized according to the literature [29]. First, 0.2 M aniline and 0.4 M formaldehyde were dissolved in toluene (50 mL), and the mixture was stirred at 40 °C for 1 h. Then, 0.2 M phenol was added into the above mixture and refluxed at 110 °C. After 6 h, the upper organic phase was transferred to a flask with one neck, and then distilled using rotary evaporators. The crude product was dissolved into anhydrous ether and washed three times with sodium hydroxide solutions (mass fraction: 5%), and then washed with distilled water to pH 7.0. The product was dried at 60 °C in a vacuum oven for 24 h and the light yellow benzoxazine was obtained.

### 2.3. Synthesis of Aniline-Cardanol Benzoxazine

The synthesis method of aniline-cardanol benzoxazine was similar to the above method. First, 0.2 M of aniline and 0.4 M of formaldehyde solution were dissolved in toluene, and the mixture was stirred

at 40 °C for 1 h. Then, 0.2 M of cardanol was added and refluxed at 110 °C for 6 h. The toluene solvent was removed using a rotary evaporator. The mixture was dissolved in anhydrous ether and washed three times with sodium hydroxide solutions, and then washed with distilled water to pH 7.0. The product was dried at 60 °C for 24 h. The resulting light orange sticky substance was obtained.

### 2.4. Synthesis of Carbon Materials

First, benzoxazine (BOZ) was dissolved in tetrahydrofuran, forming a clear and homogeneous solution. Then, SBA-15 (mass ratio, BOZ:SBA-15 = 1:1) was added into the above solution and the mixtures were sonicated for 6 h. In order to remove the tetrahydrofuran, the BOZ/SBA-15 composites were dried using a vacuum drying apparatus. After that, the BOZ/SBA-15 composites were heated stepwise at 120, 140, and 160 °C for 1 h each; at 200 and 220 °C for 2 h each; and finally at 230 °C for 1 h. Next, different carbonization temperatures were adopted. The aniline-phenol BOZ/silica composites were carbonized at 400 °C for 2 h and at 700 °C for 2 h, at 400 °C for 2 h and at 800 °C for 2 h, or at 400 °C for 2 h and at 900 °C for 2 h, with a heating rate of 2 °C/min under a high-purity nitrogen atmosphere. Furthermore, the aniline-cardanol BOZ/silica composite was carbonized at 400 °C for 2 h and at 900 °C for 2 h with a heating rate of 2 °C/min under a high-purity nitrogen atmosphere. After carbonization, 20 wt% of HF aqueous solution was used to remove the silica template. After etching, the above mixtures were centrifuged and washed with distilled water to pH 7.0. The samples were dried by vacuum drying apparatus. The as-obtained samples were denoted as NCM-700, NCM-800, NCM-900, and ACNCM-900.

### 2.5. Sample Characterization

Fourier transform infrared spectroscopy (FTIR 300, American Nicolet Co., Ltd, Waltham, MA, USA) spectra were conducted in the 400–4000 cm$^{-1}$ region. The samples were mixed with KBr powder and pressed in a pellet. In order to know the specific surface area (calculated according to the BET method), pore volume and pore size distributions (calculated by the BJH method), the nitrogen adsorption–desorption isotherms were conducted using an America Quantachrome Autosorb-iQ analyzer. The scanning electron microscopy (SEM, JSM-7500F, Osaka, Japan), transmission electron microscopy (TEM; FEI TalosF200S, Portland, OR, USA) and the X-ray diffraction (XRD, Cu Kα radiation, PANalytical B.V., Eindhoven, Netherlands) were used to characterize the morphology of carbon materials. The surface chemical species of the samples were examined using an X-ray photoelectron spectroscope (XPS; AXIS Supra, Tianjin, China).

### 2.6. Electrochemical Measurements

A three-electrode system was adopted to study the electrochemical behavior of carbon materials. In the three-electrode system, a saturated calomel electrode was the reference electrode and a platinum electrode was the counter electrode, the prepared carbon materials was the working electrode, and 0.5 M $H_2SO_4$ was the electrolyte. The preparation of the working electrode involved a carbon: acetylene black: Nafion perfluorinated resin solution at an 8:1:1 mass ratio, where the three materials were mixed in ethanol. The mass of materials coated on each working electrode was 0.2 mg. Galvanostatic charge–discharge (GCD) and cyclic voltammetry (CV) were conducted using an electrochemical workstation (CHI660E, Shanghai, China) at room temperature. The potential scan rates of cyclic voltammetry (CV) tests were 5, 10, 20, 50, and 100 mV s$^{-1}$, and the voltage window was −0.2 to 0.8 V. Electrochemical impedance spectroscopy (EIS) measurements were recorded from 10 mHz to 100 kHz. The galvanostatic charge–discharge (GCD) experiments were conducted under current densities of 0.25, 0.5, 1, 2.5, and 5 A/g. The specific capacitance (*C*, F/g) of the active material (galvanostatic discharge process) could be calculated using the following equation:

$$C = \frac{I \times \Delta t}{m \times \Delta V},$$

where $I$ (A) is the constant discharge current, $\Delta t$ (s) is the discharge time, $\Delta V$ (V) is the voltage difference during discharge, and $m$ (g) is the mass of carbon loaded into the working electrode.

In order to investigate the application of supercapacitors, a two-electrode system was also adopted. The carbon: acetylene black: poly(tetrafluoroethylene) (PTFE) binder at a 8:1:1 mass ratio was mixed. The slurry of the mixture was painted on the nickel foam (area of 1 cm$^2$) and formed a symmetrical supercapacitor device. The mass loading of the active materials on each electrode was 2.5 mg/cm$^2$. Galvanostatic charge–discharge (GCD) and cyclic voltammetry (CV) were conducted using an electrochemical workstation (CHI660E, Shanghai, China) at room temperature. The voltage window was −0.2 to 0.8 V. The electrolyte was 6 M KOH. The specific capacitance ($C$, F/g), energy density ($E$, Wh/kg) and power density ($P$, W/kg) were calculated from the discharge curves as follows [30]:

$$C = \frac{2 \times I \times \Delta t}{m \times \Delta V},$$

$$E = \frac{C \times \Delta V^2}{2 \times 4 \times 3.6},$$

$$P = \frac{3600 \times E}{\Delta t},$$

where $I$ (A) is the constant charge–discharge current, $m$ (g) is the mass of active material per electrode, $\Delta t$ (s) is the discharging time, and $\Delta V$ (V) is the potential range for the discharging process excluding the voltage drops (IR) value.

## 3. Results and Discussion

### 3.1. Microstructure Characterization

Two different benzoxazine precursors and carbon materials doped with heteroatoms were successfully synthesized. The preparation processes of the materials is illustrated in Figure 1.

The chemical structure of the BOZ monomer was confirmed using FTIR. The infrared spectrum of aniline-phenol benzoxazine is presented in Figure 2a. The bands at 1598 cm$^{-1}$ and 1489 cm$^{-1}$ corresponded to the benzene skeleton vibration. The bands at 1369 cm$^{-1}$ were assigned to the stretching mode vibration of the C–N–C group from the oxazine ring. The bands at 1226 cm$^{-1}$ and 1155 cm$^{-1}$ were assigned to the symmetric and the asymmetric stretching mode vibrations of the C–O–C group in the oxazine ring, respectively. The bands at 942 cm$^{-1}$ and 750 cm$^{-1}$ were vibrations from the oxazine ring and viscous vibration of the benzene ring, respectively.

Figure 2b gives the FTIR spectrum of aniline-cardanol benzoxazine, where the bonds at 1600 cm$^{-1}$ and 1495 cm$^{-1}$ represented the benzene ring skeleton. The bands at 1370 cm$^{-1}$ corresponded to the stretching vibration of the C–N–C in the oxazine ring. The bands at 1241 cm$^{-1}$ and 1198 cm$^{-1}$ were assigned to the symmetric and asymmetric stretching vibration of C–O–C in the oxazine ring, respectively. The peak at 955 cm$^{-1}$ was the characteristic absorption peak of the oxazine ring. The peak at 752 cm$^{-1}$ was the absorption peak of the outward viscous vibration of the benzene ring. The above characteristic peaks strongly suggested the formation of benzoxazine.

The morphology of the carbon materials was characterized using scanning electron microscopy (SEM, Figure 3a–d) and transmission electron microscopy (TEM, Figure 3e,f). Figure 3a–d are SEM images of the carbon materials (NCM-700, NCM-800, NCM-900, and ACNCM-900). The SEM images showed that the obtained N-doped carbon material had a rod-like structure. There were many channels in the rod structure. NCM-900 had a good dispersibility and uniformity due to the complete carbonization and a high nitrogen content. For the TEM images of NCM-900 and ACNCM-900 (Figure 3e,f), NCM-900 (Figure 3e) showed a more regular fringe arrangement than ACNCM-900 (Figure 3f). Within a certain range, as the nitrogen content increased, the porosity of the nitrogen-doped porous material also increased [31].

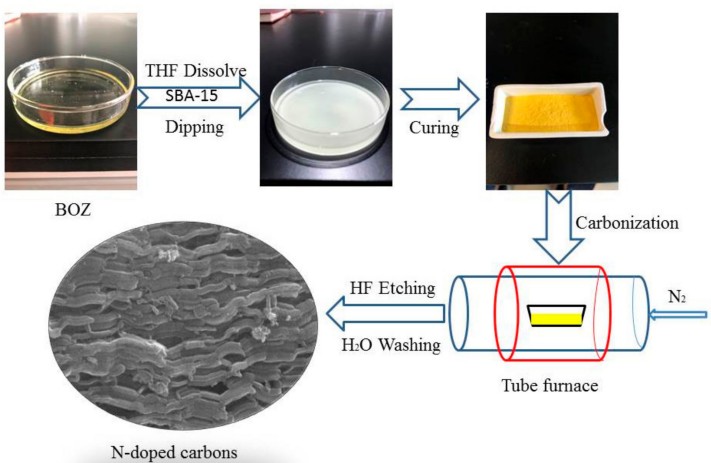

**Figure 1.** The preparation process for N-doped carbons using benzoxazine as precursor. BOZ: benzoxazine, HF: hydrofluoric acid.

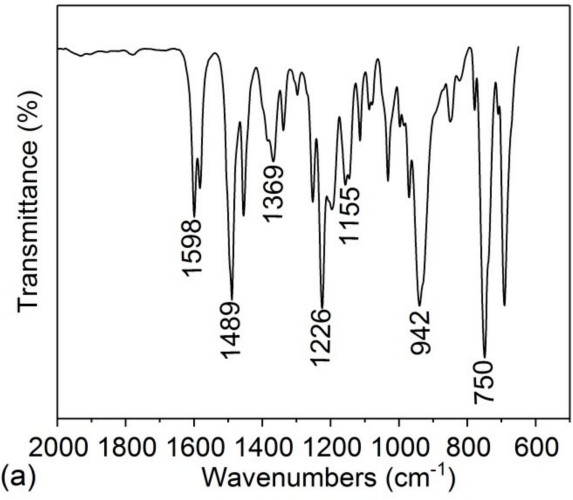

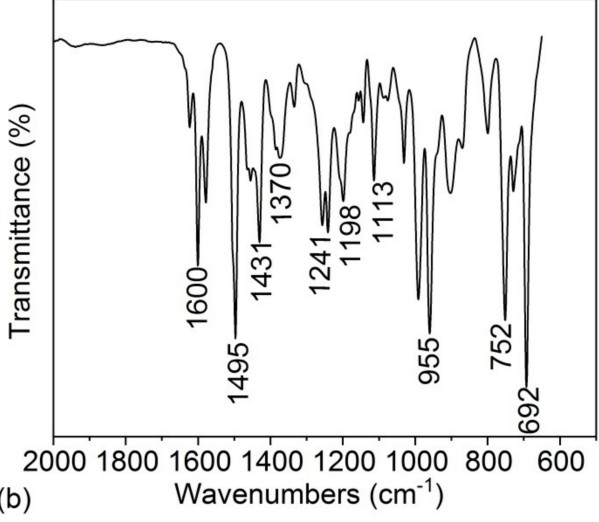

**Figure 2.** The FTIR of (**a**) aniline-phenol type benzoxazine and (**b**) aniline-cardanol type benzoxazine.

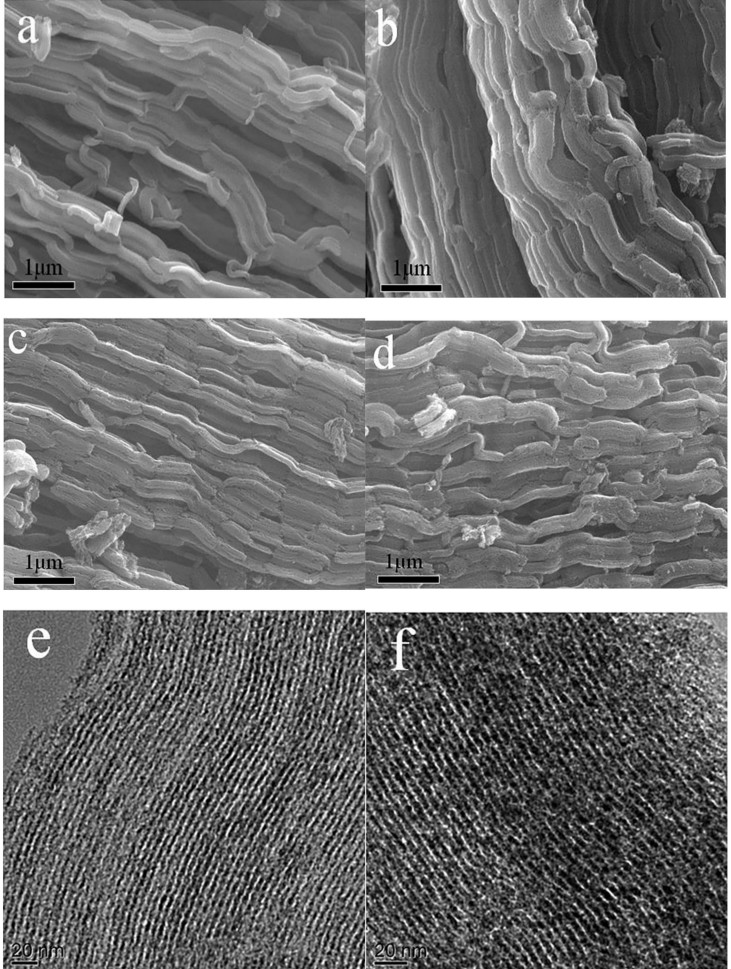

**Figure 3.** SEM images of (**a**) NCM-700, (**b**) NCM-800, (**c**) NCM-900, and (**d**) ACNCM-900; TEM images of (**e**) NCM-900 and (**f**) ACNCM-900.

The elemental compositions and nitrogen chemical states in NCM-900 and ACNCM-900 were characterized using X-ray photoelectron spectroscopy (XPS), as shown in Figure 4. The N 1s spectra of NCM-900 could be split into three peaks, corresponding to pyridinic N (397.5 eV), pyrrolic N (399.3 eV), and quaternary N (400.7 eV) (Figure 4a). The N 1s peaks of ACNCM-900 are shown in Figure 4b. It exhibited the same three nitrogen functional groups, centering at 398.1 eV, 399.3 eV, and 400.6 eV, which were also assigned to pyridinic N, pyrrolic N, and quaternary N, respectively [32]. In previous research, pyridinic N and pyrrolic N have been shown to have a positive effect at improving capacitive performance via pseudo-capacitance contribution due to configuration effects of separate electron pairs. Furthermore, the introduction of quaternary N could also facilitate electron transfer and enhance the conductivity of carbon materials [33–37]. The XPS peak analysis results of the two carbon materials are summarized in Table 1.

Small angle X-ray diffraction and wide-angle XRD were used to determine the structure of the produced carbon materials (NCM-700, NCM-800, NCM-900, and ACNCM-900) and SBA-15. The XRD patterns are presented in Figure 5a,b. The results of the XRD show that these carbon materials (NCM-700, NCM-800, NCN-900, and ACNCM-900) had an ordered and graphitized amorphous structure. The XRD also showed that these carbon materials (NCM-700, NCM-800, NCM-900, and ACNCM-900) had a similar structure to SBA-15. There were three well-resolved diffraction peaks (100, 110, and 200) in the small angle X-ray diffraction (Figure 5a), which showed that the NCM materials had a 2D hexagonal symmetry with the space group p6mm. The wide-angle XRD of NCM-900 is shown in Figure 5b, which exhibited two diffraction peaks at $2\theta = 25°$ and $44°$ belonging to the (002) and (100) planes of hexagonal

carbon, respectively [38,39]. We could find the graphitized and amorphous carbon structures of the NCM-900 from the wide angle XRD. These structures could enhance the electrochemical properties.

The nitrogen adsorption–desorption isotherms (Figure 5c) and pore size distribution (Figure 5d) indicated that these obtained carbon materials (NCM-700, NCM-800, NCM-900, and ACNCM-900) prepared using different carbonization temperatures and different precursors had typical type IV curves (according to the International Union of Pure and Applied Chemistry IUPAC classification) with a hysteresis loop (relative pressure, $P/P_0$ = 0.4–0.7) and a mesoporous structure [40].

The nitrogen adsorption–desorption isotherms of the four carbon materials were nearly similar in shape. Furthermore, compared with the curve of SBA-15 (Figure 5c), we found that the structure of the prepared carbon material came from the engraving of the template. From Figure 5d, for carbon materials prepared from the same precursor, the pore size distribution had no significant tendency even though the carbonization temperature was different. The average pore size was concentrated in the range 3.4–3.8 nm. As for the different precursors, ACNCM-900 had a larger average pore size than the NCM-900. This was because the low nitrogen content of ACNCM-900 indicated that fewer nitrogen-containing groups were carbonized during the carbonization process and fewer micropores were formed, thus the average pore size was slightly bigger. The corresponding specific surface area, average pore diameter, and total pore volume are listed in the Table 2. The specific surface areas of ACNCM-900 and NCM-900 were calculated to be 788.447 $m^2$/g and 798.623 $m^2$/g. The pore volumes of ACNCM-900 and NCM-900 were 1.376 $cm^3$/g and 1.052 $cm^3$/g. As can be seen from Table 2, the specific surface area and pore volume of carbon materials were higher than the template. The large specific surface area was beneficial for the diffusion and transportation of electrolyte ions.

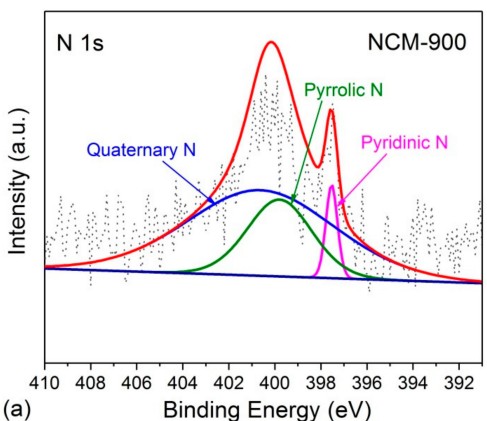

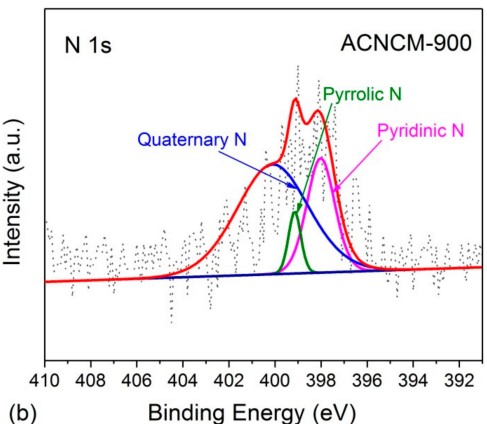

**Figure 4.** XPS spectra of nitrogen-doped mesoporous carbons (**a**) NCM-900 and (**b**) ACNCM-900.

**Table 1.** XPS peak analysis of the carbon samples.

| Samples | C (%) | O (%) | N (%) |
|---------|-------|-------|-------|
| ACNCM-900 | 93.88 | 4.64 | 1.48 |
| NCM-900 | 89.20 | 5.85 | 4.95 |

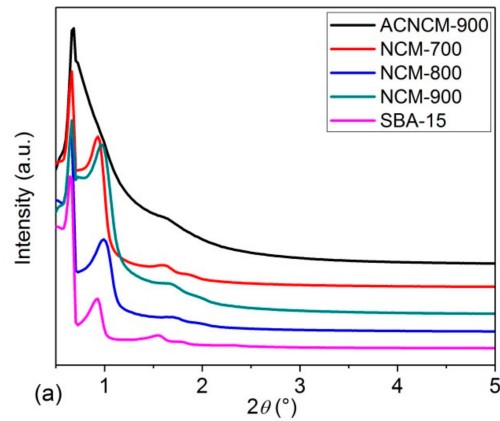

(a)

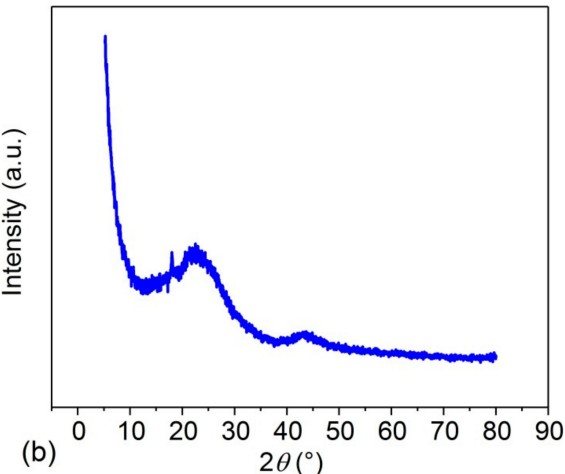

(b)

**Figure 5.** *Cont.*

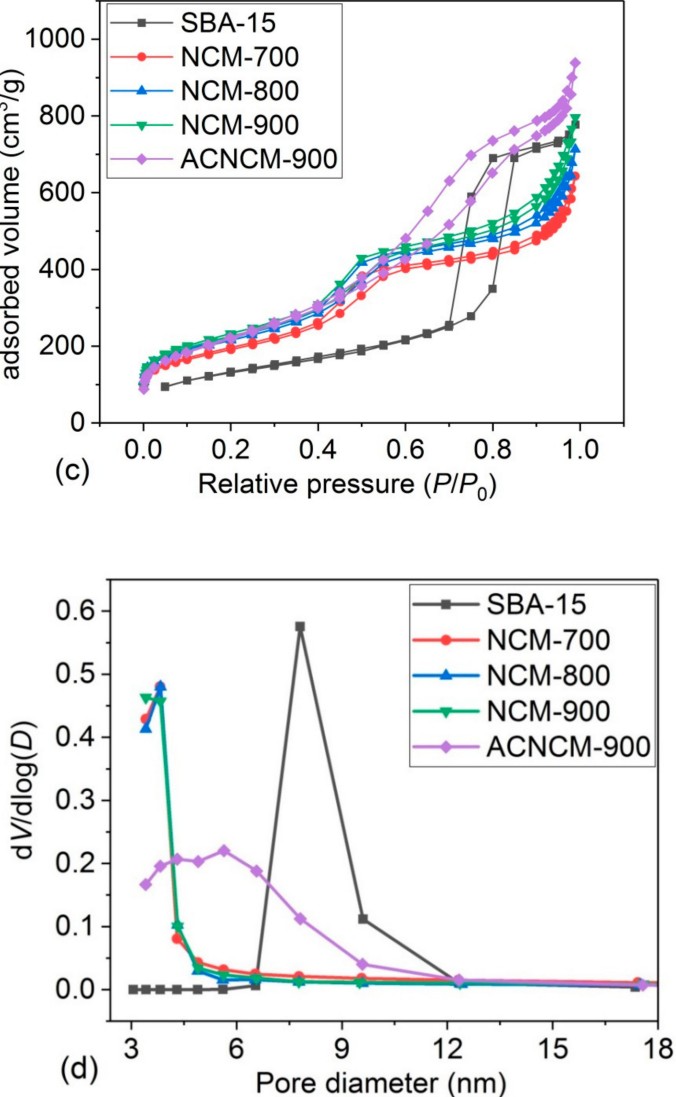

**Figure 5.** (**a**) Small angle XRD patterns of ACNCM-900, NCMs, and SBA-15; (**b**) wide angle XRD pattern of NCM-900; (**c**) nitrogen adsorption–desorption isotherms, and (**d**) pore size distribution curves of ACNCM-900, NCMs, and SBA-15.

**Table 2.** The results of nitrogen adsorption and desorption measurements.

| Samples | Specific Surface Area | Total Pore Volume | Average Pore Diameter |
|---|---|---|---|
| | (m$^2$/g) | (cm$^3$/g) | (nm) |
| ACNCM-900 | 788.447 | 1.376 | 5.639 |
| NCM-700 | 647.732 | 0.865 | 3.835 |
| NCM-800 | 759.547 | 0.934 | 3.413 |
| NCM-900 | 798.623 | 1.052 | 3.822 |
| SBA-15 | 472.677 | 1.240 | 7.810 |

*3.2. Electrochemical Performance*

In order to investigate the electrochemical performance of the above four nitrogen-doped carbon materials, cyclic voltammetry (CV) and galvanostatic charge–discharge (GCD) measurements were tested with a three-electrode system in a 0.5 M $H_2SO_4$ electrolyte at room temperature. The CV curves of the four carbon materials at the scan rate of 5 mV/s are presented in Figure 6a. It could be seen that the

curves of the four samples all presented a quasi-rectangular voltammogram shape with the redox peaks, which suggested that double-layer capacitance and pseudocapacitance characteristics were the main energy storage mechanisms of the electrode and that the redox peaks came from pseudocapacitance caused by nitrogen doping [41]. In addition, the NCM-900 had a larger area than the other materials at the rate of 5 mV/s, which indicated that the NCM-900 had a better electrochemical performance. The GCD curves of the four carbon materials at the current density of 1 A/g is presented in Figure 6b. Furthermore, the shape of the GCD curves was close to an isosceles triangle, which also indicated that the main energy storage mechanism of the electrode was double-layer capacitance. In addition, the NCM-900 had a longer discharge time than other materials at the current density of 1 A/g.

Figure 6c,d shows the cyclic voltammetry and charge–discharge curves, respectively, of NCM-900. The scan rates ranged from 5 to 100 mV/s in Figure 6c, and the shape of the curves was almost rectangular, which indicated that NCM-900 was suitable as an electrode material due to the reversible adsorption–desorption of free ions. With the scan rate increased, the curve of the sample also presented an approximate rectangular shape, which meant this material was suitable for rapid charge–discharge operation as an electrode material. Figure 6d displays the GCD curves of NCM-900 in the current density range from 0.25 to 5 A/g. The shape of the GCD curves was an isosceles triangle. At a low current density, the electrode materials exhibited the highest specific capacitance, which was contributed to by both double-layer capacitance and pseudocapacitance [26,42]. Upon an increase in current density, the pseudocapacitance gradually disappeared and the shape of isosceles triangle was maintained well, which suggested NCM-900 was a good electrode material.

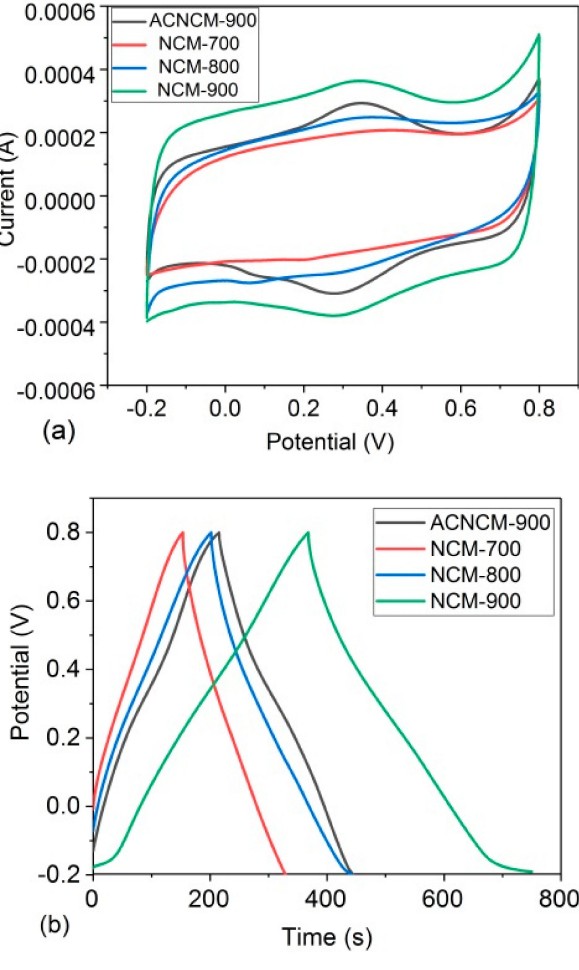

**Figure 6.** *Cont.*

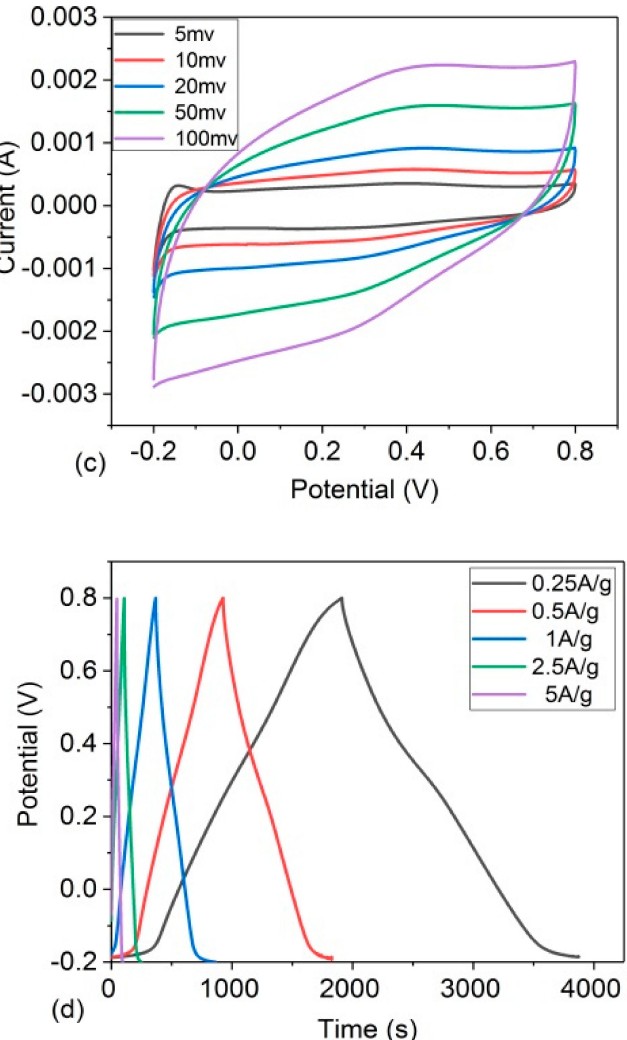

**Figure 6.** (**a**) Cyclic voltammetry (CV) curves of four carbon materials at the scan rate of 5 mV/s, (**b**) galvanic charge-discharge (GCD) curves of four carbon materials at the current density of 1 A/g, (**c**) CV curves of NCM-900, and (**d**) GCD curves of NCM-900.

In order to analyze the data intuitively, we made a comparison of the specific capacitance of different carbon materials in Table 3.

**Table 3.** The specific capacitance (F/g) of carbon materials at different current densities.

| Samples/Current Density | 0.25 A/g | 0.5 A/g | 1 A/g | 2.5 A/g | 5 A/g |
|---|---|---|---|---|---|
| NCM-700 | 236 | 205 | 178 | 139 | 96 |
| NCM-800 | 367 | 303 | 244 | 173 | 129 |
| NCM-900 | 460 | 440 | 400 | 322 | 300 |
| ACNCM-900 | 299 | 265 | 233 | 202 | 185 |

Figure 7a shows the specific capacity at different current densities of two different nitrogen-containing carbon materials. The data shows that NCM-900 had a better electrochemical performance than ACNCM-900. At the current density of 0.25 A/g, the specific capacitance of NCM-900 was 460 F/g, and when the current density was 5 A/g, the specific capacitance of NCM-900 was 300 F/g, and the capacitance retention rate was 65.2%, which is reflective of a good capacitance performance.

Figure 7b shows the cycling stability of NCM-900 and ACNCM-900 at the current density of 5 A/g. After 2000 cycles of charge and discharge, the capacitance retention rate of NCM-900 and ACNCM-900

were 96.1% and 94.3%, respectively. This suggested that these electrode materials had a good cycle stability and were suitable for use in supercapacitors.

Figure 7c shows the EIS curves of NCM-900 and ACNCM-900 in a 0.5 M $H_2SO_4$ electrolyte and the frequency range from $10^{-2}$ to $10^5$ Hz. The curve contained a semicircle and a straight line. The diameter of the semicircle at high frequencies represented the charge transfer resistance ($R_{ct}$) [43], while the linear slope of the low frequency part of the resistor characterized the diffusion resistance of the electrolyte and protons. The $R_{ct}$ of NCM-900 and ACNCM-900 were found to be 0.8 Ω and 1.1 Ω, respectively. NCM-900 had a lower resistance, which was attributed to the higher nitrogen content, which caused an increase in carbon surface polarity. This was consistent with the CV and GCD test results.

The cyclic voltammetry (CV) and galvanostatic charge–discharge (GCD) of NCM-900 in a two-electrode system are presented in Figure 8a,b, respectively. The CV curves remained as a nearly rectangular shape, which indicated ideal EDLC behavior and good reversibility for the electrolyte ions diffusing rapidly to the interface of the electrode. The GCD curves all exhibited a triangular-like shape even as the current density increased to 5 A/g, indicating the good coulombic efficiency with superior EDLC performance of NCM-900 [44].

It is confirmed by Figure 8c that NCM-900 had an excellent cycling stability in the two-electrode system and the capacitance retention was 97.5% after 2000 charge–discharge cycles at a current density of 5 A/g. The specific capacitance for NCM-900 calculated from the GCD curve is also exhibited in this graph, where a high capacitance of 317 F/g was observed at 0.25 A/g, and the specific capacitance was 200 F/g was observed when the current density was at 5 A/g.

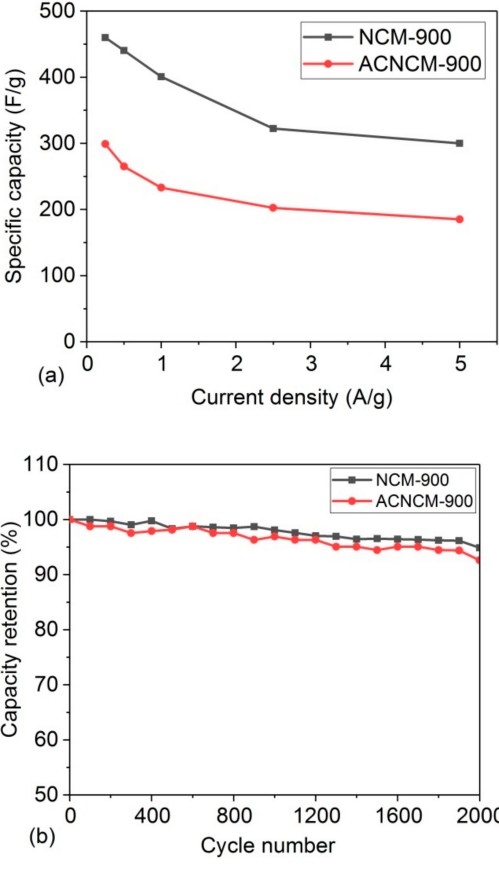

**Figure 7.** *Cont*.

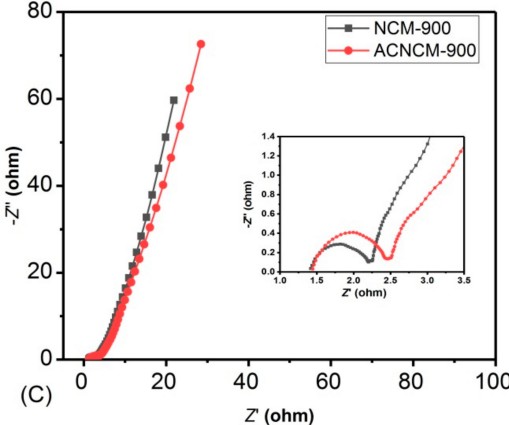

(C)

**Figure 7.** (**a**) Specific capacity of NCM-900 and ACNCM-900 at different current densities, (**b**) cycle stability of NCM-900 and ACNCM-900 after 2000 cycles at 5 A/g, and (**c**) electrochemical impedance spectroscopy (EIS) of NCM-900 and ACNCM-900.

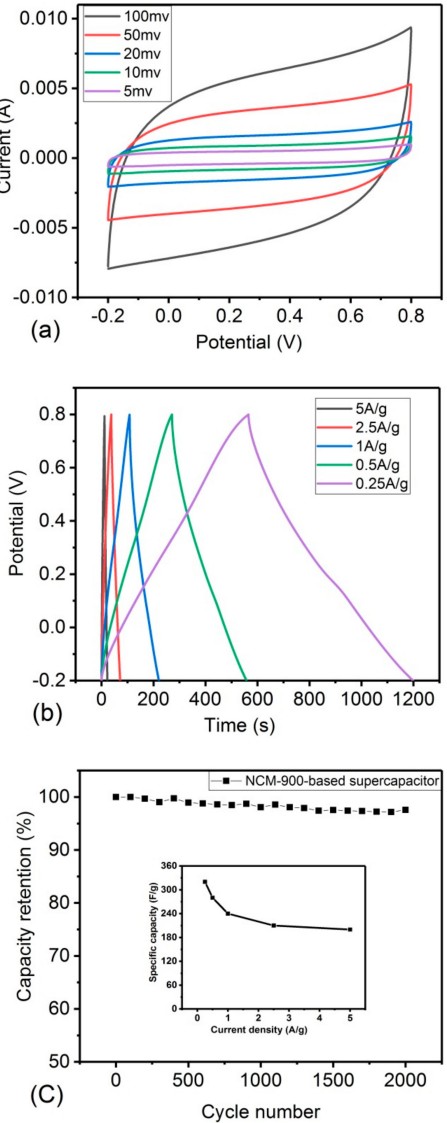

**Figure 8.** *Cont.*

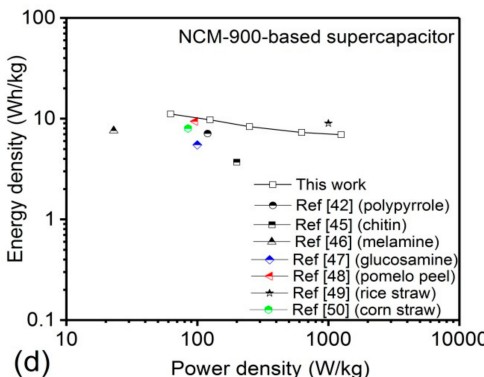

**Figure 8.** (**a**) CV curves of NCM-900 at different scan rates, (**b**) GCD curves of NCM-900 at different current densities, (**c**) cycling stability performance at a current density of 5 A/g to 2000 cycles for NCM-900-based supercapacitors in 6 M KOH electrolyte (inset graph is the specific capacity of NCM-900 at different current densities), and (**d**) Ragon plots for NCM-900-based supercapacitor with reported supercapacitors constructed from other kinds of carbon materials.

The results of the energy densities and power densities of the supercapacitors calculated for various current densities are shown as Ragone plots in Figure 8d. There was a decreasing trend in energy density with an increase of power density. The energy density of the NCM-900-based supercapacitor was calculated to be over 11.1 Wh/kg at 0.25 A/g with a power density of 62.5 W/kg. The energy density of the NCM-900-based supercapacitor was 7.1 Wh/kg at a high current density of 5 A/g with a power density of 1250 W/kg, which is highly competitive with most carbon-based materials obtained from other precursors, as shown in Figure 8d [42,45–50]. The results confirmed that N-doped carbons with an ordered pore structure and a high surface area could enable higher energy and power densities, and NCM-900 is a promising candidate for capacitive energy storage and conversion.

## 4. Conclusions

The morphology, structure, and electrochemical properties of the four samples prepared using benzoxazine were characterized, and the impact of the carbonization temperature and nitrogen content on the electrochemical properties were investigated. The NCM-900 had the highest specific capacitance of 460 F/g in a 0.5 M $H_2SO_4$ electrolyte at the current density of 0.25 A/g. At the current density of 5 A/g, its specific capacitance was still able to retain 300 F/g and the capacitance retention rate was 65.2%. Moreover, NCM-900 presented a good stability with a 96.1% capacitance retention after 2000 cycles at the current density of 5 A/g in a three-electrode system. Furthermore, the high specific capacitance, comparable energy/power densities, and excellent cycling stability of NCM-900 were exhibited using a symmetrical electrode system. Therefore, NCM-900 is a promising candidate for high-performance supercapacitors.

**Author Contributions:** This paper was accomplished based on collaborative work of the authors. H.Z. performed the experiments, analyzed the data, interpreted the experimental results and wrote the paper. L.X. contributed to the experimental design. L.X. and G.L. supervised the entire research progress. All authors have read and agreed to the published version of the manuscript.

**Funding:** This research was funded by the National Natural Science Foundation of China grant number (51703205).

**Conflicts of Interest:** The authors declare no conflict of interest.

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
