# Peer review of "Synthesis of Benzoxazine-Based N-Doped Mesoporous Carbons as High-Performance Electrode Materials"

_applsci, doi:10.3390/app10010422_

Round 1
Reviewer 1 Report
The manuscript on nitrogen-doped carbons from benzoxazine as electrode materials is just mere extention of previous reports. This reviewer did not see any new innovation in this work.
The quality of the manuscript can be improved with the following.
1) Introduction part is misleading, it clearly said the post treatment of carbons with acids will decrease in surface area and are unstable, and in the presented work they use HF to clean the materials.
In fact HF is more bad than any others listed in the introduction.
A good re-write is needed.
2) Use of SBA-15 is not clear. This reviewer is not convinced on the use of SBA-15. Rather this reviewer recommend the authors to think of any sacrificial template that can offer better carbon yields and improved surface area and porosity or us a template that does not need HF.
3)Manuscript at the current condition is not suitable to be accepted.
Reviewer 2 Report
This manuscript suggests n-doped carbon for enhancing electrochemical performances. The results were interesting. However, some questions have to be solved to publish Applied Sciences.
Authors selected SBA 15 as template. Do authors have any particular reason to choose SBA 15? In Table 2, pore size does not show any tendency unlike specific surface area and total pore volume. More explanations are needed. In figure 7 (b), capacitances were fluctuated during cycling, this cycle-ability behavior is not normal. Thus, authors have to explain about these results. Generally, imaginary value (y-axis) of the Nyquist plot reversely proportional to capacitance (R_im=1/(2*pi*f*C)). The ACNCM-900 showed higher capacitance than that of the NCM 800. This result is not matched with Figure 7 (a). It is recommended to specify the loading amount of the electrode. I recommend to add energy densities and power densities (the Ragone plot) to compare the characteristics with other studies.Author Response
Please see the attachment.

Round 2
Reviewer 2 Report
The previous comments are well addressed, and the manuscript is acceptable now.
Author Response
Dear Reviewer,
Thank you for your comments concerning our manuscript entitled “Synthesis of benzoxazine based N-doped mesoporous carbons as high performance electrode materials (ID: applsci-655496)”. And sincerely thank you for your hard work on this paper.
Best regards!
Haihan Zhang
